# Effects of Broccoli Rotation on Soil Microbial Community Structure and Physicochemical Properties in Continuous Melon Cropping

Xiaodi Liu [1], Xuelian Ren [1], Shuangshuang Tang [1], Zhaoran Zhang [1], Yufei Huang [1], Yanqiu Sun [1], Zenggui Gao [1,*] and Zhoujie Ma [2,*]

1  College of Plant Protection, Shenyang Agricultural University, Shenyang 110866, China; xiaodiliu@163.com (X.L.); renxuelian1753@163.com (X.R.); tangtss@foxmail.com (S.T.); zhangzhaoran1986@foxmail.com (Z.Z.); huangyufei@syau.edu.cn (Y.H.); syqiu77@163.com (Y.S.)
2  Maize Research Institute, Shanxi Agricultural University, Xinzhou 034000, China
*  Correspondence: gaozenggui@outlook.com (Z.G.); snmzj123@163.com (Z.M.)

**Abstract:** The limitations and weaknesses of continuous melon cropping have worsened in recent years. A melon–broccoli rotation can possibly alleviate the problems associated with melon monoculture; however, the underlying mechanisms and their impact on the rhizosphere's soil microbial community remain unclear. Thus, high-throughput sequencing was used to evaluate the rhizosphere soil's microbial community's relative abundance and diversity under melon–broccoli rotation and continuous melon monoculture cropping systems. We found that relative fungal and bacterial diversity and richness increased while fungi relative abundances, such as *Fusarium* spp. were significantly decreased under broccoli rotation. During continuous cropping, enriched Acidobacteria and *Streptomyces* spp., *Sphingomonas* spp., and *Pseudomonas* spp. were identified, which play important roles in alleviating melon continuous cropping obstacles. The soil under continuous cropping was rendered acidic, underwent secondary salinization, rapidly accumulated soil organic carbon and nitrogen, and lost abundant phosphorus and potassium. In contrast, broccoli rotation partially mitigated these negative physicochemical responses. Redundancy analysis revealed that the soil pH, soil soluble salt content, and soil organic carbon were linked to structures of the soil bacterial and fungal community. Melon–broccoli rotation could effectively equilibrate the soil microenvironment and overcome the challenges and deficiencies associated with continuous melon cropping.

**Keywords:** broccoli; continuous melon cropping; crop rotation; physicochemical properties; microbial community



## 1. Introduction

Melon (*Cucumis melo*) is an economically vital fruit crop and the second most commercially important member of the Cucurbitaceae (FAOSTAT, http://faostat.fao.org accessed on 1 September 2022) [1]. China began cultivating melons >3000 years ago. Melon has been widely cultivated in China and in temperate tropical regions worldwide. The rapid development of the melon industry has increased revenue for Chinese farmers. The recent increase in the demand for melons and the accelerated development of greenhouse agriculture have made greenhouse cultivation the main method of melon production [2]. However, greenhouse cultivation has also popularized continuous cropping, which has substantially decreased the melon yield by hindering fruit growth and development [3–5]. A stable soil microbial community is vital to crop productivity as it establishes and maintains good soil health [6]. The rhizosphere is complex and changeable. Continuous cropping can result in the deterioration of rhizosphere physicochemical properties [7], the accumulation of autotoxic substances [8–10], and changes in soil microbial community diversity [11,12]. Microorganisms are the most active soil components and enable soils to provide ecosystem

services by participating in biogeochemical cycling [13,14] and maintaining soil health and quality [15]. Soil microbial characteristics substantially affect the impediments related to continuous cropping [11,16]. Continuous cropping obstacles are critically influenced by changes in the soil microbial composition. Continuous cropping creates imbalances in rhizospheric microorganisms, increases the abundance of pathogenic fungi, decreases the abundance of certain beneficial fungi, weakens plant growth, and significantly increases disease incidence [17–19]. It also reduces several microbial species and the organic matter content in the soil and increases the incidence of soilborne disease [20]. The soil environment and cultivation practices affect soilborne disease incidence and severity, especially in greenhouse-raised crops such as cucumbers, tomatoes, and peppers [21–23].

Soil physicochemistry affects the soil microbial community's composition. Changes in resident soil microflora are associated with alterations in soil physicochemical properties [24]. Changes in soil pH may dramatically change the abundance of soil bacterial flora and exacerbate continuous cropping obstacles [25]. The soil's pH is a strong predictor of soil fungal community composition [26]. It also determines the abundance and activity of certain soil bacteria [27] and influences soil bacterial and archaeal community composition [28]. Soil microorganisms determine soil biochemical and physicochemical properties and are, therefore, vital indicators of soil fertility and productivity [29,30]. Nevertheless, other factors, such as soil nitrogen and phosphorus content, may directly or indirectly influence the soil microbial community's distribution [31–33]. Nitrogen content also affects soil microbial growth, community type, and number [34,35]. In addition, soil salinity influences rhizosphere microbial composition [36,37] and is a major factor in shaping soil bacterial diversity in many natural habitats [38]. Crop rotation alleviates continuous cropping obstacles [39,40]. Crop rotation influences soil function and crop productivity and reduces the incidence and severity of adverse soil-related factors in modern intensive agriculture systems [41,42]. Hence, crop rotation has gained increasing attention as an important disease management tool in sustainable agriculture. It lowers the incidence of soilborne pathogens, maintains soil fertility, and indirectly influences the soil microbiota by modulating soil structure and physicochemistry [43]; the deterioration of soil physicochemical properties is known to worsen continuous cropping outcomes [44]. In contrast, crop rotation protects production by altering the soil microenvironment. It can effectively improve soil physicochemistry, regulate soil fertility, and increase crop yield [45]. For example, the metabolites and products of root stubble decomposition of rape, green onion, and pepper can inhibit the growth of the fungal pathogen *Fusarium oxysporum* while promoting melon root growth; thus, rape, green onion, and pepper are planted before melon [46]. Rotational strip maize–peanut intercropping has increased peanut yield and quality and has overcome other impediments in relation to continuous peanut cropping [47]. Ding et al. [48] have shown that green garlic/cucumber intercropping can sustainably and efficiently enhance cucumber production and somewhat improve the soil environment. Hong discovered that pepper–banana and eggplant–banana rotations resulted in a significant decrease in pathogen abundance [49]. Li's study [50] suggested that rotation was beneficial to the improvement of soil ecosystem versatility and the development of disease-suppressing soil compared with potato continuous cropping. Therefore, the soil-crop-microbial ecological system could be stabilized using crop rotation to attenuate continuous cropping obstacles.

Cruciferous plants such as broccoli have received extensive agricultural research attention [51]. Glucosinolate is a secondary metabolite, and its by-product, isothiocyanate (ITC), inhibits a variety of soilborne pathogens such as *Fusarium* spp. and *Rhizoctonia solani* [52,53]. Broccoli (*Brassica oleracea* L. *var. italica* L.) reduced the number of *Verticillium* dahliae Kleb and microsclerotia in the soil and the incidence of *Verticillium* wilt in cauliflower (*Brassica oleracea* L. *var. botrytis* L.) [54–56]. Wang et al. [57] reported that the number of microsclerotinia in the soil significantly decreased after broccoli rotation and broccoli residue amendment compared with continuous potato cropping. Hence, broccoli rotation could be used to control potato *Verticillium* wilt. A broccoli–strawberry rotation system is an economical and feasible option for the augmentation of strawberry growth and

yield [55]. Based on the foregoing results, we hypothesize that broccoli rotation could also be effective against melon wilt. Preliminary laboratory studies have shown that rotation with crucifers such as broccoli and rape can promote melon growth, lower the incidence of melon wilt, and allelopathically inhibit *Fusarium oxysporum*. However, the changes in soil's physicochemistry and the microbial community abundance and composition that occur during melon–broccoli rotation are unknown. We compared the physicochemical properties of soils from fields wherein (a) melons had never been planted, (b) melons were continuously planted, and (c) broccoli was planted after continuous melon cropping. We analyzed soil bacterial and fungal community diversity and composition using high-throughput sequencing. We also investigated the sources of the obstacles related to continuous cropping and the microecological changes that occurred after crop rotation to provide rational scientific recommendations for the sustainable development of melon production. We made two hypotheses: (1) continuous melon cropping could destroy the balance of the soil microbial community's structure and increase the abundance of harmful fungi in the soil, and (2) broccoli rotation could improve the physicochemical properties of the rhizosphere soil under continuous melon cropping, balancing the soil's nutrient content.

## 2. Materials and Methods

### 2.1. Study Sites and Soil Sample Collection

The tested muskmelon variety was named "Kaqi Green Gem" (produced by Shen He Seed Industry Co., Ltd., Cangzhou, China). The tested broccoli variety was named "Green Sword" (produced by Geng Yun Seed Industry Co., Ltd., Tianjin, China). The experiment was conducted in a glass greenhouse in Shenyang County, Liaoning Province, China (123°34′16″ E, 41°49′53″ N). This region has a temperate, semi-humid, continental monsoon climate, with an average annual temperature of 8 °C and an average annual precipitation of 714 mm. The soil used for the experiment was taken from a fallow site where tomatoes were planted before the fallow. The collected soil was mixed and evenly distributed into a 15 cm × 20 cm pot with 3 kg of soil per pot.

First of all, sampling was taken randomly at 5 points and then at a total of 15 points (that is, three repetitions). These time-start samples without experimental treatment (March 2019) could be used as the control group (CK) for samples in subsequent experiments. From March 2019 to March 2020, a continuous cropping experiment was conducted, with melon planted three times during this period. The rotation experiment then started in April 2020 and lasted until March 2021. All management measures of each treatment were consistent. Melon continuous cropping soil samples (M) were collected in March 2020, and the broccoli rotation samples (MB) were collected in March 2021. The rhizosphere soil samples were collected via a five-point method. After removing 0–5 cm of the surface soil and gently shaking off the soil around the root system, the soil that adhered to the root surface was brushed off for collection. Each soil sample was separated into two subsamples by passing it through a <2 mm sieve. One was used to evaluate the soil's physicochemical properties, while the other was used immediately for total genomic soil DNA extraction and then Illumina sequencing. Each treatment had three repetitions.

### 2.2. Determination of Soil Physicochemical Properties

The rhizosphere soil physicochemical properties were analyzed according to previously reported methods [58]. The soil pH was determined using the potentiometric method (soil:water = 1:2.5). The soil soluble salt (SS) content was determined using the electrode method (soil:water = 1:5). This was based on a standard conductivity curve, and the concentration at room temperature, using the soil electrical conductivity.

The potassium dichromate volumetric-external heating method was used to determine the soil organic carbon (SOC) content. The sodium hydroxide melting, molybdenum antimony sulfate colorimetry, flame photometry, and sulfuric acid digestion Kjeldahl methods were used to quantify total phosphorus (TP), potassium (TK), and nitrogen (TN), respectively. The available nitrogen (Ava-N), nitrate nitrogen ($NO_3{}^-$-N), ammonium

nitrogen ($NH_4^+$-N), available phosphorus (Ava-P), and available potassium (Ava-K) were determined using the alkaline hydrolysis diffusion, potassium chloride soaking dual-wavelength colorimetry, indophenol blue, sodium bicarbonate extraction molybdenum antimony colorimetry, and ammonium acetate soaking atomic absorption spectrometry methods, respectively.

### 2.3. Soil DNA Extraction and Illumina Sequencing

Th total genomic soil DNA was extracted with an E.Z.N. ATM Mag-Bind soil DNA kit (Omega Bio-Tek, Norcross, GA, USA) according to the manufacturer's protocol. DNA concentration and purity were determined on 1% (*w/v*) agarose gels. The primers 515F (5′-GTGCCAGCMGCCGCGG-3′) and 907R (5′-CCGTCAATTC-MTTTRAGTTT-3′) were used to amplify the V4–V5 hypervariable region of the 16S rRNA gene. The fungal ITS region was amplified via PCR using the ITS1F (5′-CTTGGTCATTTAGAGGAAGTAA-3′) and ITS1R (5′-GCTGCGTTCTTC-ATCGATGC-3′) primer pairs. The 30 μL reaction system consisted of a 15 μL 2× Phanta Master Mix, 1 μL primer F, 1 μL primer R, 10–20 ng DNA, and dd $H_2O$, adding up to 30 μL. The PCR reaction program was pre-denatured at 98 °C for 5 min with denaturation at 98 °C for 30 s, annealing at 55 °C for 30 s, and extension at 72 °C for 45 s, for a total of 27 cycles, and a final extension at 72 °C for 10 min. The PCR reaction was carried out on an ABI GeneAmp® 9700 (Applied Biosystems Trading (Shanghai) Co., Ltd., Shanghai, China). The PCR product was electrophoresed on 2% (*w/v*) agarose gel. The mixture was blended to the same concentration as the PCR product. The blended PCR products were purified with an AxyPrep DNA gel extraction kit (Axygen Scientific Inc., Union City, CA, USA), electrophoresed on 2% (*w/v*) agarose gel, and quantitated with QuantiFluor™-ST (Promega, Madison, WI, USA). The Illumina Nova-Seq PE250 platform (Illumina, San Diego, CA, USA) was used to sequence the constructed library, and 250-bp paired-end reads were generated. Sequencing was performed at Shenyang Boster Biotechnology Co. Ltd., Shenyang, China.

### 2.4. Data Processing and Statistical Analysis

Samples were clustered with Usearch v. 7.0.1090 (https://www.drive5.com/usearch/ (accessed on 15 December 2021)). The sequences were divided into different OTUs based on ≥97% similarity. Representative sequences were obtained based on OTU clustering. The most abundant sequence was selected to represent the OTU and was subjected to various OTU analyses. Mothur v. 1.41.0 (https://github.com/mothur/mothur/releases/ (accessed on 15 December 2021)) was used to calculate the alpha diversity indices, including Chao1, Shannon, Simpson, and Coverage. Rarefaction and rank abundance curves were plotted to evaluate the sequencing reliability and uniformity coefficients. Beta diversity analyses were performed to compare the differences in the microbial community structure among samples using UniFrac distance metrics [59]. Beta diversity was visualized using PCoA, NMDS, and UPGMA hierarchical clustering (R v. 3.4.4; R Core Team, Vienna, Austria). Taxonomy was assigned based on the naïve Bayesian RDP classifier v. 2.2 (https://sourceforge.net/projects/rdp-classifier/ (accessed on 15 December 2021)) using 0.7 as the minimum confidence threshold to assign a taxonomic classification level to each OTU with ≥97% similarity [60]. Classification results below the threshold were designated as "unclassified" [61].

The soil physicochemical properties and microbiological index data were analyzed using SPSS Statistics v. 19 (IBM Corp., Armonk, NY, USA). Data are the mean ± standard error of the mean (SEM) of at least triplicate experiments. They were subjected to one-way ANOVA and Fisher's least significant difference (LSD) test. Differences were considered significant at $p < 0.05$ (*) and highly significant at $p < 0.01$ (**). Pearson's correlation coefficient analyses and redundancy analyses (RDA) were run in Canoco for Windows v. 4.5 (Biometris, Wageningen, The Netherlands) to identify relationships between the soil environmental factors and the soil microbial community diversity.

## 3. Results

### 3.1. Soil Physicochemical Properties

The changes in the soil sample's physicochemical properties are listed in Table 1. The soil pH decreased with continuous melon cropping but increased after broccoli rotation. Nevertheless, the direction of the changes in the soil soluble salt content (SS) and soil organic carbon content (SOC) was the opposite of that for the soil pH. The total nitrogen (TN) content was higher for group M than for groups CK and MB. However, the total phosphorus (TP) and total potassium (TK) were lowest for group M. On the other hand, the observed change in TK was not significant. Available nitrogen (Ava-N), nitrate nitrogen ($NO_3^-$-N), and ammonium nitrogen ($NH_4^+$-N) were enriched in the soil under continuous melon cropping. In contrast, AK and AP were substantially lower under continuous melon cropping than under broccoli rotation.

**Table 1.** Physicochemical properties of rhizosphere soil.

| Indicator | CK | M | MB |
|---|---|---|---|
| pH | $6.61 \pm 0.06$ [a] | $5.34 \pm 0.01$ [b] | $6.84 \pm 0.01$ [c] |
| SS ($g \cdot kg^{-1}$) | $2.14 \pm 0.05$ [b] | $3.74 \pm 0.03$ [a] | $1.48 \pm 0.27$ [c] |
| SOC ($g \cdot kg^{-1}$) | $20.34 \pm 0.26$ [b] | $25.83 \pm 2.98$ [a] | $18.05 \pm 0.47$ [b] |
| TN ($g \cdot kg^{-1}$) | $2.76 \pm 0.08$ [ab] | $2.95 \pm 0.07$ [a] | $2.62 \pm 0.16$ [b] |
| TP ($g \cdot kg^{-1}$) | $2.27 \pm 0.07$ [a] | $1.58 \pm 0.06$ [b] | $1.73 \pm 0.05$ [c] |
| TK ($g \cdot kg^{-1}$) | $17.98 \pm 0.30$ [a] | $16.74 \pm 0.67$ [a] | $17.92 \pm 0.59$ [a] |
| Ava-N ($mg \cdot kg^{-1}$) | $185.72 \pm 1.36$ [b] | $198.41 \pm 5.41$ [a] | $150.08 \pm 2.53$ [c] |
| $NO_3^-$-N ($mg \cdot kg^{-1}$) | $7.83 \pm 1.92$ [b] | $15.47 \pm 0.68$ [a] | $3.03 \pm 2.38$ [c] |
| $NH_4^+$-N ($mg \cdot kg^{-1}$) | $13.03 \pm 0.52$ [c] | $92.59 \pm 20.10$ [a] | $51.77 \pm 3.23$ [b] |
| Ava-P ($mg \cdot kg^{-1}$) | $236.02 \pm 1.71$ [a] | $142.16 \pm 16.89$ [b] | $232.60 \pm 1.99$ [a] |
| Ava-K ($mg \cdot kg^{-1}$) | $549.18 \pm 3.22$ [b] | $336.34 \pm 3.24$ [c] | $929.96 \pm 3.42$ [a] |

Index data were analyzed using SPSS Statistics v. 19. The value is the mean ± standard error of three repeated experiments (*n* = 3). Different letters indicate significant differences ($p < 0.05$). CK, control; M, fields where melons had been continuously planted; MB, fields where broccoli had been planted after continuous cropping of melons. SS, soluble salt; SOC, soil organic carbon; TN, total nitrogen; TP, total phosphorus; TK, total potassium; Ava-N, available nitrogen; $NO_3^-$-N, nitrate nitrogen; $NH_4^+$-N, ammonium nitrogen; Ava-P, available phosphorus; Ava-K, available potassium.

### 3.2. Soil Microbial Community Alpha Diversity

The operational taxonomic unit (out) rarefaction and rank abundance curves confirmed the generation of adequate sequencing data (Figure S1). The sample's relative abundance (observed species and Chao1 index) and diversity (Shannon and Simpson indices) are shown in Table 2. Continuous melon cropping increased the bacterial Chao1 and Shannon alpha diversity indices. Relative soil bacterial richness increased and alpha diversity decreased after broccoli rotation; however, the differences were not significant. Fungi in treatments MB and control CK had the highest and second highest abundances and diversity, respectively. Compared with the control, fungal richness and diversity were lower after continuous melon cropping and higher after broccoli rotation. Hence, continuous cropping reduced the fungal species diversity and distributed the fungal species unevenly.

**Table 2.** Soil bacterial and fungal richness and diversity.

| Index | Bacteria | | | Fungi | | |
|---|---|---|---|---|---|---|
| | CK | M | MB | CK | M | MB |
| Chao1 | $2798.78 \pm 224.69$ [b] | $3754.67 \pm 170.48$ [a] | $3874.76 \pm 42.19$ [a] | $589.19 \pm 315.92$ [b] | $327.19 \pm 105.40$ [b] | $1133.84 \pm 58.84$ [a] |
| Observed species | $2456.33 \pm 202.92$ [b] | $3466.00 \pm 165.30$ [a] | $3495.00 \pm 17.06$ [a] | $556.33 \pm 290.89$ [b] | $305.67 \pm 95.87$ [b] | $1076.00 \pm 62.45$ [a] |
| Shannon | $7.84 \pm 0.56$ [b] | $9.59 \pm 0.14$ [a] | $9.57 \pm 0.03$ [a] | $6.02 \pm 0.78$ [a] | $3.85 \pm 0.85$ [b] | $6.18 \pm 0.31$ [a] |
| Simpson | $0.97 \pm 0.01$ [a] | $1.00 \pm 0.00$ [b] | $1.00 \pm 0.00$ [b] | $0.96 \pm 0.02$ [a] | $0.8178 \pm 0.0766$ [b] | $0.94 \pm 0.01$ [a] |

Data were analyzed using SPSS Statistics v. 19. The value is the mean ± standard error of three repeated experiments (*n* = 3). Different letters show significant differences ($p < 0.05$).

### 3.3. Composition of Soil Microbial Communities

A total of thirty-four distinct bacterial phyla were detected across all nine samples. The most abundant sequences were affiliated with the phyla Proteobacteria (25.36–52.55%) and Actinobacteria (12.61–22.73%) followed by Chloroflexi (5.97–14.34%), Firmicutes (7.61–11.55%), Bacteroidetes (3.64–9.23%), Patescibacteria (3.08–6.94%), Acidobacteria (4.15–4.48%), Planctomycetes (2.90–6.10%), Gemmatimonadetes, (1.77–6.15%), and Verrucomicrobia (0.39–1.16%) (Figure 1A). The order of Betaproteobacteriales in the Proteobacteria (25.32% in group CK) predominated. *Xanthomonas* in the Proteobacteria (17.32% and 12.91% in groups CK and MB, respectively) was the second most dominant. The orders Clostridiales (6.14%), Sphingomonadales (5.60%), Bacteroidales (6.07%), and Streptomycetales (5.40%) predominated in group M and were more abundant under this treatment than all others (Figure 1C). Ten distinct fungal phyla were detected across all nine samples, and Ascomycota predominated (40.37–66.65%). Basidiomycota and Mortierellomycota represented only 0.64–5.48% and 0.64–2.40% of all the reads, respectively (Figure 1B). The abundance of Hypocreales in the CK group was only 12.07% but increased to 45.44% after continuous melon cropping and decreased to less than that of the CK group (namely, 9.99%) after broccoli rotation. Eurotiales and Sordariales were comparatively more abundant, whereas Capnodiales, Mortierellales, Agaricales, Glomerellales, Sporidiobolales, Microascales, Cantharellales, Onygenales, Archaeorhizomycetales, Tremellales, Chaetothyriales, Pleosporales, Helotiales, Saccharomycetales, and Pezizales had the lowest richness in group M (Figure 1D).

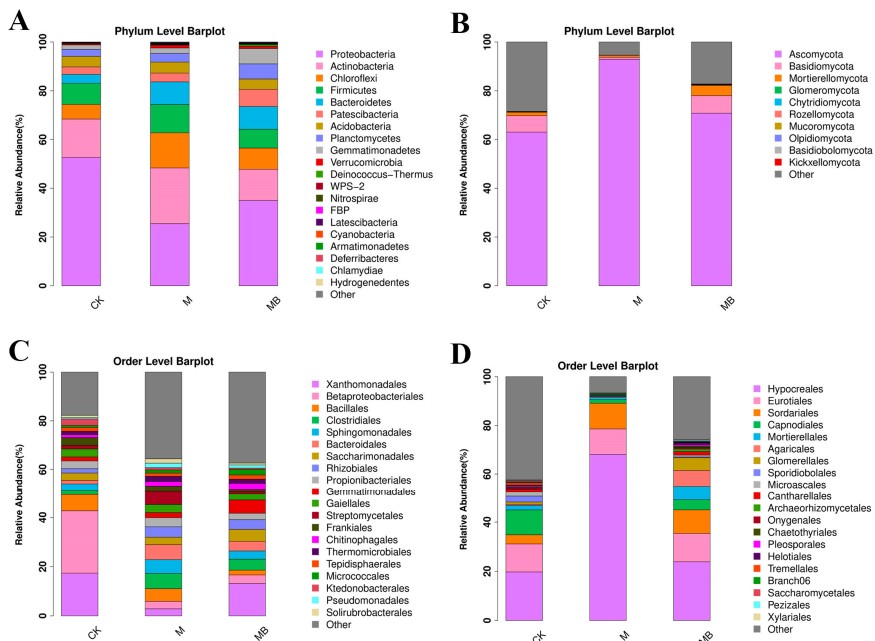

**Figure 1.** Bar plot analysis for relative abundance at the (**A**) Bacterial phylum, (**B**) Fungal phylum, (**C**) Bacterial order, and (**D**) Fungal order levels, showing the composition and structure of soil from fields where melons had never been planted (group CK), fields where melons had been continuously planted (group M), and fields where broccoli had been planted after the continuous cropping of melons (group MB).

The relative abundances of *Sphingomonas* spp., *Lachnospiraceae NK4A136* group, *Marmoricola* spp., *Nitrolancea* spp., *Pseudomonas* spp., and *Ruminococcaceae UCG-014* were nonsignificantly higher in group M than the others. However, the abundances of *Streptomyces* spp., *Pseudolabrys* spp., and *Altererythrobacter* spp. significantly decreased after broccoli rotation. Nevertheless, *Chujaibacter* spp. and *Mizugakiibacter* spp. significantly decreased under continuous melon cropping but were restored to half their abundance levels in group CK after broccoli rotation (Figure 2A, Table S1). Continuous melon increased the abundance

of *Fusarium* spp. The latter accounted for 37.31% of all fungal sequences. On the contrary, *Fusarium* spp. abundance decreased after broccoli rotation and was restored to a level below that of group CK. *Coniochaeta* spp. abundance varied in the same manner as *Fusarium*. The abundances of ten of the top twenty fungal genera were lower under continuous melon cropping but higher under broccoli rotation. *Trichothecium* spp. and *Rhinocladiella* spp. were detected only in group MB. *Penicillium* and *Conlarium* non-significantly decreased in response to the treatments (Figure 2B).

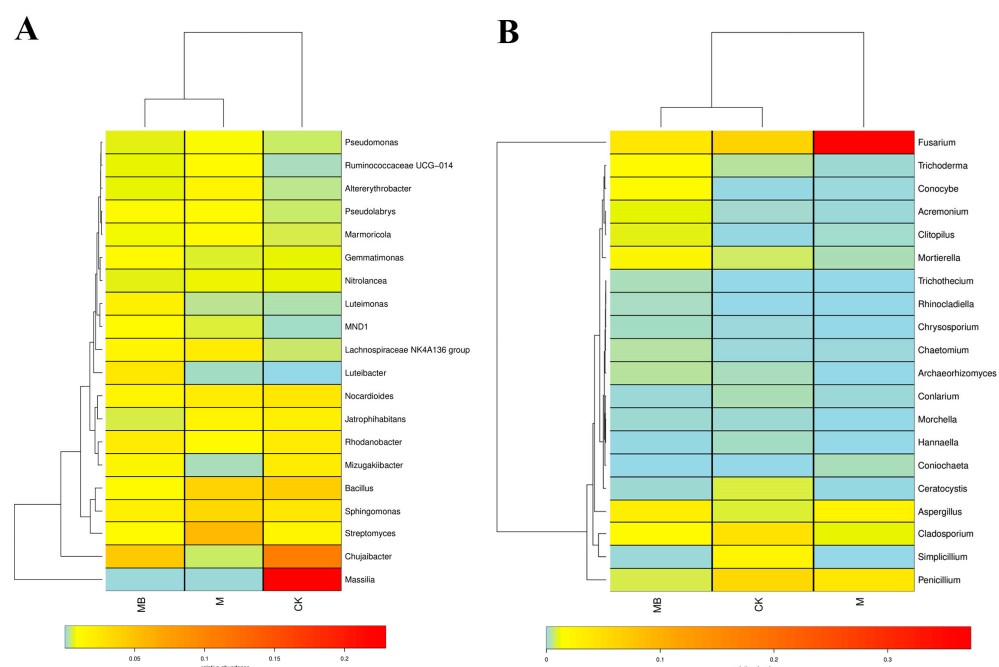

**Figure 2.** Heatmap analysis of the top 20 bacteria (**A**) and fungi (**B**) from the soil of three groups at the genus level. The relative abundance for each genus is depicted using the color intensity in each region. Higher relative abundance values are represented using red, whereas lower relative abundance values are represented using blue. Treatment abbreviations are defined in Figure 1.

*3.4. Soil Microbial Community Beta Diversity Analysis*

The UniFrac distances revealed the significant effects of crop management on the soil microbial community. The multiple response permutation procedure (MRPP) analyses of the 16S rRNA gene and ITS data according to the unweighted and weighted methods were performed, showing that the bacterial and fungal community structures and compositions changed significantly and the intergroup differences were greater than the intragroup differences (Table 3).

**Table 3.** MRPP analysis of bacteria and fungi based on Unifrac.

| | Unweighted | | | Weighted | | |
|---|---|---|---|---|---|---|
| | Observed Delta | Expected Delta | Pr (>F) | Observed Delta | Expected Delta | Pr (>F) |
| Bacteria | 0.2770 | 0.4043 | 0.005 ** | 0.2427 | 0.4284 | 0.003 ** |
| Fungi | 0.5295 | 0.6441 | 0.008 ** | 0.3566 | 0.5961 | 0.006 ** |

** indicates significant differences ($p < 0.01$). The smaller the observed delta value, the smaller the difference within the group. The larger the expected delta value, the larger the difference between groups.

Hierarchical clustering and principal coordinate analyses (PCoAs) based on the UniFrac distances showed differences in the soil microbial community composition between the groups. A soil cluster tree diagram (Figure 3A,B) showed that the bacterial and fungal communities in the soil samples from groups M and MB were divided into the first

group. Hence, these soil samples differed from those of group CK. The M and MB samples had relatively greater bacterial and fungi community structure similarity (Figure 3A,B). The PCoA (Figure 3C,D) revealed that the distribution of the soil samples among different treatments was relatively independent. Thus, the soil community structures differed. PCoA ordination disclosed variations between the groups for bacterial communities. PC1 (first principal component; 41.61% contribution) and PC2 (second principal component; 22.1% contribution) differentiated in the bacterial communities (Figure 3C). PC1 (35.39% contribution) and PC2 (16.6% contribution) differentiated the fungal communities (Figure 3D). The same phenomena were observed through the PCoA and NMDS analyses based on Bray–Curtis distances (Figure S2).

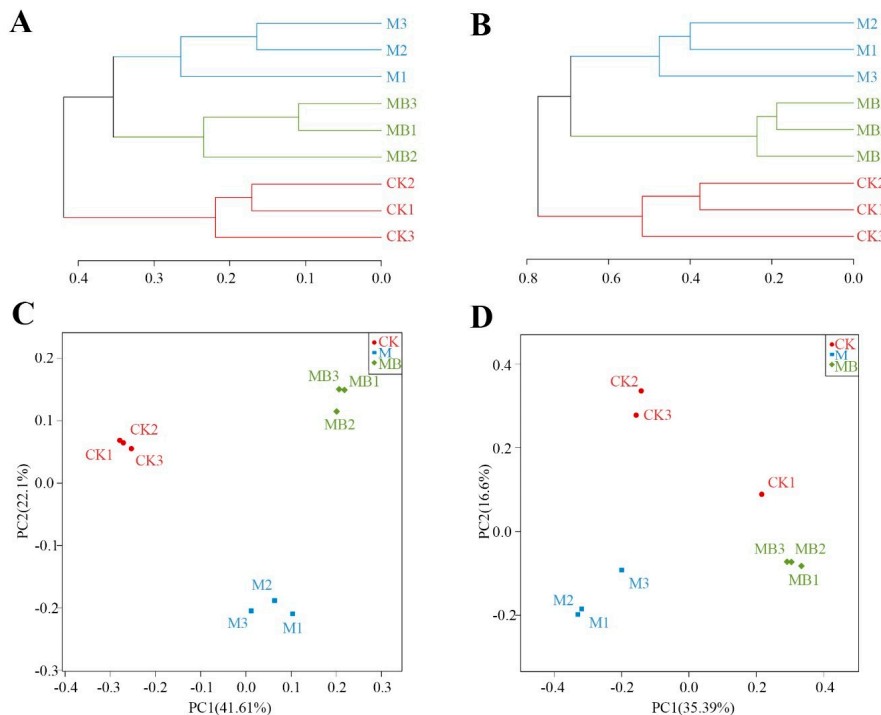

**Figure 3.** Clustering analysis of bacteria (**A**) and fungi (**B**) for 9 soil samples based on a weighted unifrac algorithm. Principal coordinate analysis of bacteria (**C**) and fungi (**D**) for 9 soil samples based on unweighted unifrac algorithm.

*3.5. Analysis of Correlations among Soil Environmental Factors and Soil Microbial Communities*

A redundancy analysis (RDA) suggested that changes in the soil chemistry shaped the soil microbial community composition. For each treatment matrix, two axes explained 93.6% and 98.3% of the total bacterial variability, respectively (Figure 4A,B), and 99.7% and 99.8% of the total fungal variability, respectively (Figure 4C,D).

The changes in pH, SS, SOC, and TP (Figure 4A), as well as Ava-N, $NH_4^+$-N, Ava-K, and Ava-P (Figure 4B), strongly influenced the alteration in the soil bacterial community's structure and composition, whereas the changes in TK, TN, and $NO_3^-$-N had a relatively low impact on these parameters. The changes in pH, SS, and SOC (Figure 4C), as well as Ava-P, Ava-N, and Ava-K (Figure 4D), strongly affected the soil fungal community structure and composition. Nevertheless, the changes in TP, TK, TN, $NO_3^-$-N, and $NH_4^+$-N had a comparatively lower impact on these parameters. Thus, pH, SS, and SOC exerted the greatest influence on rhizosphere microorganisms.

The top five bacterial phyla, in terms of abundance, exhibited different degrees of correlation with the changes in the soil's environmental factors. Proteobacteria were strongly and positively correlated with pH, TP, TK, Ava-P, and Ava-K. Actinobacteria and Chloroflexi were strongly and negatively correlated with these indices. Proteobacteria were strongly and negatively correlated with TN, SS, SOC, $NO_3^-$-N, and $NH_4^+$-N. Actinobacteria

and Chloroflexi were strongly and positively correlated with these indices. Firmicutes and Bacteroidetes were strongly and positively correlated with soil nitrogen and strongly and negatively correlated with soil potassium and phosphorus. At the fungal phylum level, Ascomycota were negatively correlated with pH, TP, TK, Ava-P, and Ava-K but positively correlated with SOC, SS, TN, Ava-N, $NO_3^-$-N, and $NH_4^+$-N. Basidiomycota, Glomeromycota, and Chytridiomycota were negatively correlated with SOC, SS, TN, and $NH_4^+$-N but positively correlated with pH, TP, TK, and Ava-P.

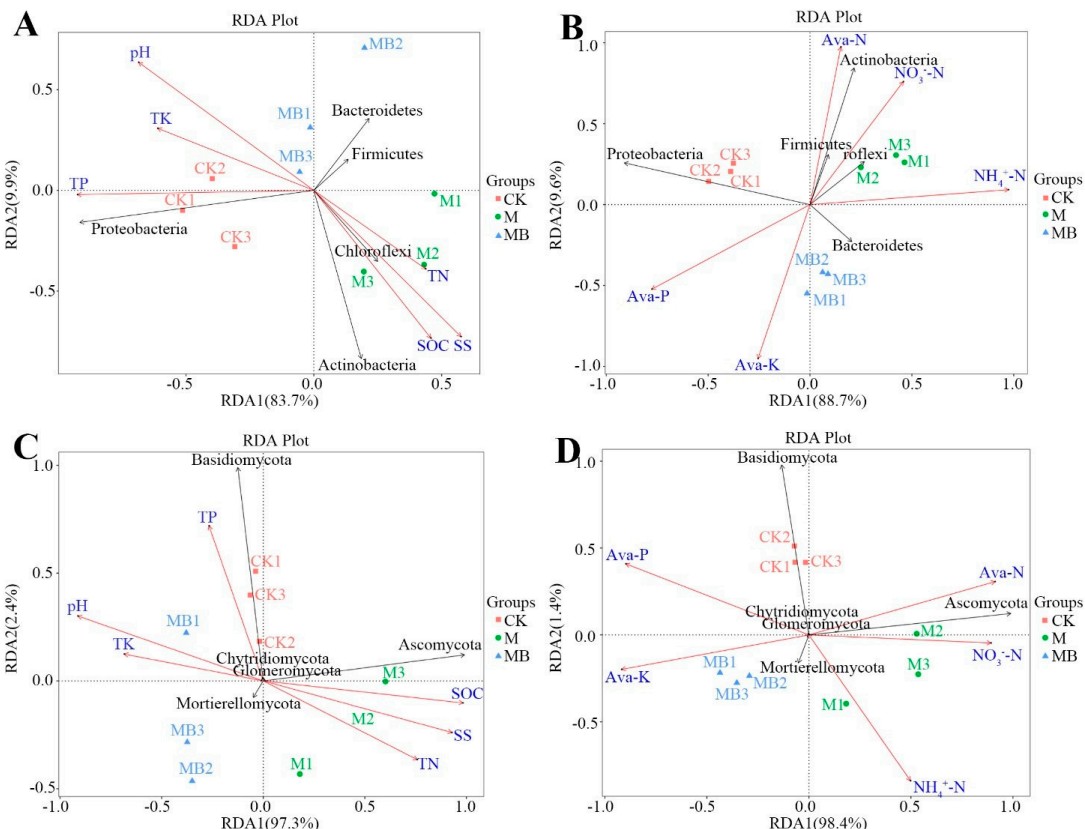

**Figure 4.** Correlations between soil environmental factors with bacteria (**A**,**B**) and fungi (**C**,**D**). Red arrows represent the physical and chemical properties of the soil; black arrows represent the soil microorganisms (phylum level). The angle between the influencing factors (between the factors and the microorganisms) indicates the positive correlation (acute angle) and a negative correlation (obtuse angle) between two factors; the arrow length represents the intensity of the effect for each factor. SS, soluble salt; pH, matrix pH; SOC, soil organic carbon; TN, total nitrogen; TP, total phosphorus; TK, total potassium; Ava-P, available phosphorus; Ava-K, available potassium; Ava-N, alkali nitrogen; $NO_3^-$-N, nitrate nitrogen; $NH_4^+$-N, ammonium nitrogen. Treatment abbreviations are defined in Figure 1.

Spearman's correlation analyses of the soil environmental factors and the top 10 microbial phyla returned results that were similar to those of the RDA (Figure S3A and Figure 3B). Spearman's rank correlations were used to evaluate the relationships between soil physicochemistry and the top 10 bacterial and fungal genera. *Massilia* spp. were positively correlated with all indices except TN, Ava-K, and $NH_4^+$-N and were significantly positively correlated with TP ($p < 0.05$). *Bacillus* spp., *Streptomyces* spp., and *Sphingomonas* spp. were strongly positively correlated with SS ($p < 0.05$, $p < 0.001$, ns), SOC ($p < 0.05$, $p < 0.01$, $p < 0.05$), and $NO_3^-$-N ($p < 0.05$, $p < 0.05$, $p < 0.05$) and negatively correlated with pH ($p < 0.05$, $p < 0.05$, ns). *Chujaibacter* spp. was strongly and negatively correlated with SOC ($p < 0.05$), TN ($p < 0.01$), $NO_3^-$-N ($p < 0.05$), and $NH_4^+$-N ($p < 0.05$) and negatively correlated with TP ($p < 0.05$) and AP ($p < 0.05$) (Figure S3C). Of the top 20 fungal genera, *Fusarium* spp.

was the most significantly increased in group M relative to group CK. *Fusarium* spp. were positively correlated with SS ($p < 0.01$), SOC ($p < 0.01$), TN (ns), Ava-N ($p < 0.01$), $NO_3^--N$ ($p < 0.05$), and $NH_4^+-N$ (ns). Conversely, all fungal genera that were significantly decreased in group M compared with group CK were positively correlated with pH, TP, TK, Ava-P, and Ava-K to varying degrees (Figure S3D).

## 4. Discussion

### 4.1. Changes in Soil Physicochemical Properties

Continuous cropping was found to influence the soil's pH and nutrient levels. Soil pH significantly decreased with the prolongation of continuous cropping and affected the crop incidence rate [62]. Long-term continuous melon cropping lowered the soil's AK and AP content, raised the soil's AN content, and caused nutrient imbalances [63]. The soil's SS, pH, and nutrient content are critical factors in continuous melon cropping. Obstacles to continuous melon cropping include decreases in soil pH and phosphorus and potassium content and increases in SS and nitrogen content, which are caused by long-term monoculture. Here, the soil pH of group M was considerably lower than that of group CK and increased after rotation (Table 1). This finding was consistent with those of previous studies [63,64] and suggests that a decrease in soil pH is a major obstacle to continuous melon cropping. The soil's pH affects all soil physicochemical and biological properties [65]. The soil's pH was negatively correlated with TN, AN, $NO_3^--N$, and $NH_4^+-N$ (Table S2). These results are consistent with those of earlier works demonstrating that soil pH can be associated with changes in the quantities of $NH_4^+$ and $NO_3^-$ released into the soil [66,67]. Our discoveries revealed that TN, AN, $NO_3^--N$, and $NH_4^+-N$ significantly increased in response to continuous melon cropping but were restored after broccoli replanting (Table 1). Organic nitrogen mineralization increases the soil pH by consuming $H^+$ during ammonification and then releasing H+ during nitrification [68]. Here, we found that the soil pH was negatively associated with $NO_3^--N$. We also observed that the soil nitrogen content was significantly reduced after broccoli rotation. Liu et al. [69] proposed that the co-application of $NO_3^--N–NH_4^+-N$ at appropriate ratios could improve broccoli productivity, quality, and economic return. Therefore, changes in TN, $NO_3^--N$, and $NH_4^+-N$ could modulate the obstacles related to continuous melon cropping.

Available P and K decreased under continuous cropping systems [58]. In contrast, crop rotation and intercropping alleviated the soil nutrient imbalances and decreases caused by continuous single cropping [70]. Lyu [40] showed that Chinese cabbage or bean rotation could alter substrate physicochemical properties and significantly improve the total K and available P content under continuous tomato cropping. Similarly, we discovered that soil AK and AP were significantly higher under broccoli rotation than continuous melon cropping (Table 1). Thus, broccoli rotation could change soil physicochemical properties and significantly improve the soil phosphorus and potassium content under continuous melon cropping.

The soil organic matter and C content are key drivers and indices of soil health. They play central roles in soil fertility, function, quality, and sustainability [71]. Chen et al. [72] showed that the soil organic matter content increased with the prolongation of continuous passionfruit cropping. The same results were reported for a study on rhizosphere soil physicochemical properties across several planting years [73]. We found that the soil organic carbon content ($25.83 \ g \cdot kg^{-1}$) significantly increased under continuous melon cropping compared to CK ($20.34 \ g \cdot kg^{-1}$). This result was consistent with that reported by Yuan et al. [64]. The relative effects of continuous cropping on the soil organic content can vary with soil and plant types. Miao et al. [74] demonstrated that fallow soil had a relatively higher organic carbon content than under continuous corn cropping. Börjesson [75] reported significantly higher soil organic carbon content under a ley-dominated rotation than monoculture cereal cropping. It is nonetheless difficult to establish the impact of continuous cropping on soil organic matter and carbon content because these metrics can widely vary among soils and crop systems. However, crop root systems, plant metabolites,

and soil biodiversity could determine the effects of continuous cropping on soil organic matter content [38].

Here, SS was remarkably higher for group M than group CK. Nevertheless, the SS was restored to the level below that of group CK under broccoli rotation. Hence, increases in the soil salt content associated with continuous melon cropping might, in fact, be reversible. Numerous earlier works illustrated that soil salt content is vital to plant growth [76,77]. This result indicates that developing effective methods of reducing soil salinity after melon planting could help alleviate the continuous cropping problem.

*4.2. Changes in Soil Microbial Community Composition*

For this study, broccoli rotation improved the relative bacterial richness in the rhizosphere soil. Li et al. [78] found that if pathogens were suppressed, the bacterial richness in the soil increased. Their results indicated that a more abundant microbial community could better protect the host from pathogens. This was more pronounced in soils where *Fusarium* disease was suppressed compared to soils where *Fusarium* disease was severe [79]. Several studies have examined the soil microbial bacterial community structure under continuous cropping systems [80,81]. Many bacterial taxa have proven beneficial to crops in terms of conferring disease protection, augmenting plant growth, and inducing systemic pathogen resistance [82–84]. Will [85] reported that *Chloroflexi* relatively increased in nutrient-poor soil layers and was negatively correlated with soil nitrate concentration. *Chloroflexi* can degrade organic residues [86]. Shen [79] found that changes in Acidobacteria and Bacteroidetes abundance were linked to disease suppression. Here, we found that Actinobacteriota, Chloroflexi, Firmicutes, Bacteroidetes, and Acidobacteriota were more abundant in soil under continuous melon cropping than they were under crop rotation. These results are similar to those of Tan [17], who reported that certain bacterial phyla, such as Acidobacteria, Actinobacteria, and Cyanobacteria, were comparatively more abundant under continuous cropping systems. We also discovered that the relative abundances of *Streptomyces* spp., *Sphingomonas* spp., and *Pseudomonas* spp. were significantly higher than those of other genera in soils under continuous cropping. Nevertheless, the relative abundances of *Rhodanobacter* spp. and *Gemmatimonas* spp. decreased in soils under continuous cropping (Figure 2A, Table S1). *Pseudomonas* spp. suppress various microbial pathogens such as *Thielaviopsis basicola* [87]. *Pseudomonas* spp. have been utilized in the biocontrol of banana *Fusarium* wilt disease. In field conditions, banana *Fusarium* wilt has been controlled up to 79% using *Pseudomonas* spp. strains [88,89]. Shen et al. [79] suggested that *Pseudomonas* spp. and *Acidobacteria* might play an important role in the natural suppression of banana *Fusarium* wilt disease. *Streptomyces* spp. are promising as biocontrol agents because they are antagonistic to various plant pathogens [90]. *Sphingomonas* spp. promote plant growth under various stress conditions [91]. *Gemmatimonas* spp. can help regulate phosphate metabolism and stabilize the soil microbial community [92], while *Rhodanobacter* spp. are halosensitive Gram-negative aerobic bacteria [93]. In summary, these studies indicated that *Acidobacteria*, *Streptomyces*, *Sphingomonas*, and *Pseudomonas* could facilitate and improve continuous melon cropping via its antagonism toward harmful microorganisms.

Fungi are key components of the microbial community, affecting soil stability and fertility and exerting various ecological functions such as decomposition, parasitism, pathogenesis, and symbiosis [94]. In the intensive vegetable production system of China, the effects of long-term continuous cropping on soil quality and especially on the genetic and functional diversity of fungi have not been widely examined [40]. Continuous cropping can alter the soil fungal community's structure. Here, PCoA revealed the clear separation of soil fungal communities under continuous melon cropping when subjected to broccoli rotation (Figure 3B). This finding was confirmed through hierarchical clustering (Figure 3D). There were significant differences in the soil fungal genetic communities under various cultivation techniques. These discrepancies accounted for the observed differences in the fungal community structure between continuous in monoculture and rotation cropping [95,96]. Soil fungal richness and diversity were lower under crop rotation than under continuous

monocrop cultivation [97]. Cabbage and bean rotation reduced fungal richness and diversity under continuous tomato cropping [40]. Gao et al. [18] reported that soil fungal diversity and richness were significantly increased in soil under continuous cropping. On the contrary, Dong [98] found that the diversity of soil fungi significantly decreased after three years of continuous *Panax notoginseng* cropping. Continuous cropping may reduce the diversity and alter the composition of the soil fungal community. Continuous soybean cropping can cause root rot by increasing soil *Fusarium* abundance [99]. Soil's total microbial community diversity was higher under wheat–soybean rotation than monoculture [100]. Lyu et al. [40] found that celery rotation improved soil fungal richness and diversity compared with continuous cropping. Wang et al. [101] showed that fungal diversity decreased with increasing the duration of cultivation. Decreases in fungal diversity can increase the risks and death rates associated with plant disease. In contrast, increased soil fungal diversity might actually ameliorate plant pathogenesis. Here, we discovered that broccoli rotation increased the soil fungal community's diversity. This finding was consistent with a prior study suggesting that crop rotation increased microbial community diversity compared to monoculture [102]. Another investigation revealed that crop rotation lowered the abundance of certain plant pathogens while enhancing the abundance of certain plant-growth-promoting microorganisms [103]. Hence, there were disagreements among the foregoing studies concerning the effects of different cropping systems on soil fungal diversity and, by extension, plant disease suppression. Nevertheless, the preceding investigations suggested that the fungal species composition rather than diversity could be important for soil health. It is necessary to further research and elucidate the roles of fungal community parameters in the emergence and suppression of soilborne diseases under a wide range of soil and crop types [104].

Fungi decompose soil organic matter and play vital roles in terrestrial ecosystems [105]. Basidiomycota, Chytridiomycota, and Zygomycota are the main fungal phyla implicated in this process [106]. Compared with continuous melon cropping (M), broccoli rotation (MB) decreased the relative abundances of Ascomycota while increasing the relative abundances of Basidiomycota, Mortierellomycota, Glomeromycota, and Chytridiomycota. These findings were consistent with those for celery–tomato rotation [40]. Qin et al. [107] found that the relative abundances of Zygomycota and Basidiomycota were higher for continuous potato cropping under ridge and furrow mulching cultivation than those for continuous potato cropping under flat ridge mulching cultivation.

Crop rotation also significantly lowered the abundances of the fungal pathogens *Fusarium* and *Coniochaeta* (Table S1). *Fusarium* spp. are well-known pathogens of watermelon, banana, cucumber, and tomato [108]. Continuous soybean cropping increased *Fusarium* spp. abundance and susceptibility to root rot [99]. Relative to vanilla monoculture, there was a significantly lower abundance of *Fusarium oxysporum* in the vanilla rhizosphere under black-pepper–vanilla rotation [109]. Certain *Coniochaeta* spp. are plant pathogens that might use lytic polysaccharide monooxygenases (LPMOs) as pathogenicity factors. Similar findings were reported for the maize pathogen *Colletotrichum graminicola* [110]. We found that *Mortierella* and *Acremonium* were beneficial to plant growth and positively correlated with crop rotation (Table S1). *Mortierella* can generate antagonistic arachidonic acid, which elicits phytoalexins and suppresses plant disease. Hence, *Mortierella* can help maintain the microecological balance by inhibiting soilborne pathogens [111]. *Mortierella* can also solubilize phosphatic rocks and increase soil P bioavailability [112]. Grunewaldt-Stöcker and von Alten [113] demonstrated that root pre-inoculation with the endophytes *Acremonium* significantly lowered disease incidence by triggering systemic resistance in the host plant. *Acremonium* might also be antagonistic to *Fusarium* through hyphal interactions. The former might suppress root rot disease and *Fusarium* wilt in tomatoes and melons, respectively [114]. Here, continuous melon cropping decreased the abundance of antagonistic fungi, increased the abundance of pathogenic fungi, enriched pathotrophs, and augmented soilborne disease rates. However, broccoli rotation alleviated soilborne diseases that are associated with continuous melon cropping.

*4.3. Relationship between the Microbial Community and Soil Physicochemistry*

Crop rotation can effectively protect and enhance overall vegetable production. It changes the soil microenvironment, improves soil physicochemistry, regulates soil fertility, and increases crop yield [45]. Here, we used RDA plots and Spearman's correlation analyses to demonstrate the interactions among soil physicochemical properties and microbial communities. The most abundant bacterial taxa that were negatively regulated using crop rotation were negatively correlated with soil pH, total K and P, and available K and P (Figure S3A,C). These findings were consistent with the fact that crop rotation negatively regulated the most abundant fungal taxa (Figure S3B,D). The results of the present study are aligned with those of previous works showing that soil pH is vital to microbial community construction [115,116]. Acidobacteria have a wide range of metabolic and genetic functions. The soil pH influences the Acidobacteria's community composition and structure [117]. Acidobacteria displayed robust inverse responses to soil pH [24]. The relative abundance of Acidobacteria increased with decreasing soil pH [116,118]. The RDA demonstrated that Acidobacteria were strongly negatively correlated with soil pH. However, the observed increase in Acidobacteria abundance in M soil with a high SOC content contradicts the previously reported finding that Acidobacteria prefers nutrient-poor soils [103]. This discrepancy could be explained by the fact that Acidobacteria has diverse subdivisions with unique soil nutrient requirements [119]. Acidobacteria are generally sensitive to changes in the soil pH. We found that pH strongly influenced the rhizosphere soil's fungal community composition. Similar results were reported in several earlier studies. Fungi usually have narrow ranges of pH tolerance, and changes in the soil pH directly influence the fungal community composition. In the present study, the abundances of Ascomycota, Mortierellomycota, and *Fusarium* significantly changed with soil pH (Figure S3B,D). A previous study [120] reported that nutrient status and the AP, SS, $NH_4^+$-N, and $NO_3^-$-N levels are important drivers of soil fungal community composition. We observed that SS, Ava-N, $NH_4^+$-N, and $NO_3^-$-N were correlated with the soil microbial community structure to varying degrees. The discoveries made herein are consistent with those of earlier works and demonstrate that soil physicochemistry has a strong impact on soil microbial community composition [116,121,122].

## 5. Conclusions

The present study examined the effects of broccoli rotation on the physicochemical properties and the microbial community structure, abundance, and diversity of the rhizosphere soil under continuous melon cropping. Broccoli rotation improved the physicochemical properties of the rhizosphere soil under continuous melon cropping, balanced the soil nutrient content, and reduced the soil salt content. It also increased the relative abundance of bacterial richness and fungal diversity while decreasing the relative abundance of fungi in the soil that are harmful to melons. Hence, the results indicate the possibility that broccoli rotation can equilibrate the soil microbial ecological environment and somewhat overcome the obstacles associated with continuous melon cropping.

**Supplementary Materials:** The following supporting information can be downloaded at: https://www.mdpi.com/article/10.3390/agronomy13082066/s1, Figure S1. Rarefaction curve (A,B) and rank abundance curve (C,D) of bacterial (A,C) and fungal (B,D) OTU of 9 soil samples; Figure S2. Principal coordinate analysis (PCoA) (A,B) and Nonmetric multidimensional scaling analysis (NMDS) (C,D) based on Bray-Curtis for bacterial (A,C) and fungal (B,D) community composition in the five soil groups; Figure S3. Spearman's correlation analyses between the soil properties and abundance of the top 10 bacterial phyla (A), fungal phyla (B), bacterial genera (C) and fungal genera (D); Table S1. The relative abundance (%) of top 20 bacteria and fungi from the soil of three groups at the genus level; Table S2. Spearson's correlation coefficients between soil physicochemical properties.

**Author Contributions:** Conceptualization, X.L. and Z.G.; methodology, X.L., X.R. and S.T.; software, X.L. and Z.Z.; validation, Y.H., Y.S. and Z.M.; formal analysis, X.R.; investigation, Z.Z.; resources, Z.G.; data curation, X.L., X.R. and S.T.; writing—original draft preparation, X.L.; writing—review and editing, Z.G. and Z.M.; visualization, X.L.; supervision, Z.G.; project administration, Z.G.; funding acquisition, Z.G. All authors have read and agreed to the published version of the manuscript.

**Funding:** This research was funded by the Independent Innovation Fund for Agricultural Science and Technology of Ningxia Hui Autonomous Region (NGSB-2021-10-01) and the Special Fund for Agro-scientific Research in the Public Interest (201503110).

**Data Availability Statement:** The data presented in this study are available on request from the corresponding authors.

**Acknowledgments:** We would like to thank all the staff at the greenhouse of the College of Plant Protection for their help with muskmelon planting and Shenyang Boster Biotechnolog Co., Ltd. (Shenyang, China) for their high-throughput sequencing and bioinformatics analysis support.

**Conflicts of Interest:** The authors declare no conflict of interest.

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
