# Peer review of "Effects of Broccoli Rotation on Soil Microbial Community Structure and Physicochemical Properties in Continuous Melon Cropping"

_agronomy, doi:10.3390/agronomy13082066_

Round 1
Reviewer 1 Report
The Materials and Methods section has many shortcomings that render this research paper unpublishable in current form. There are vague descriptions of methodologies, which makes it impossible to evaluate scientific merit of the methodologies employed in the study. This also makes the study non repeatable/reproducible. The authors need to answer the following questions as they revise the paper:
(1) What were the conditions (temperature and humidity) in the glass greenhouse where the crops (melons and broccoli) were grown?
(2) What was the biomass yield of the crops - and how was harvesting done? Was any of the crop residue biomass incorporated into the soil after each harvest cycle, since the crops were planted three times?
(3) How was it possible to fit 3 crop cycles of water melon into 12 months? At what growth stage was the melon terminated?
(4) What was the type of soil used in this study - soil classification and texture?
(5) How much soil was used in each pot? What was the size of the pots and how many melon plants were in each pot? This off course is very important to know as it can be directly related to the root biomass/concentration in the pot.
(6) Were any ammendments applied .e.g fertilizers and what rate was used?
(7) How was irrigation water managed?
8) How was harvesting done?
9) What cultivars were used for the melons and broccoli? Could it be possible that certain microbial communities can be favoured by certain cultivars? hence its important to disclose this. There are just too many cultivars of melons out there..
10) What was the experimental design? At what level of the experiment was replication of treatments done?
(11) In most cases, it is only the heads of broccoli harvested and the leaves are incorporated into the soil. Was this the case in this study?
(2)
The English language should be revised. There are some statements that may have grammar errors, hence meaning is distorted. For example, in the methodology section, the meaning of the following statement is not clear " ..The collected soil was mixed and evenly distributed into pots, and samples were taken randomly at 5 points sampling as a sample immediately, taken a total of 15 points"..
Author Response
Response to Reviewer 1 Comments
Dear Reviewer:
Thank you very much for your constructive comments and your hard work in my manuscript before. This time, I revised my manuscript carefully and made some improvements to my English writing. I really appreciate all your comments and suggestions.
(1) What were the conditions (temperature and humidity) in the glass greenhouse where the crops (melons and broccoli) were grown?
Response 1: Temperature:25-32℃; humidity:50%-60%.
(2) What was the biomass yield of the crops - and how was harvesting done? Was any of the crop residue biomass incorporated into the soil after each harvest cycle, since the crops were planted three times?
Response 2: We removed crop residues after the melons were planted 60 days.
(3) How was it possible to fit 3 crop cycles of water melon into 12 months? At what growth stage was the melon terminated?
Response 3: The growth cycle of melon is 80-90 days. Terminated after melon planted 60 days.
(4) What was the type of soil used in this study - soil classification and texture?
Response 4: We used brown soil type.
(5) How much soil was used in each pot? What was the size of the pots and how many melon plants were in each pot? This off course is very important to know as it can be directly related to the root biomass/concentration in the pot.
Response 5: The collected soil was mixed and evenly distributed into a 15 cm×20 cm pot, the amount of soil in each pot was 3 kg. One plant was retained in each pot.
(6) Were any ammendments applied .e.g fertilizers and what rate was used?
Response 6: We added 15 kg of organic fertilizer (N, P, and K≥ 12 g/kg) before the soil was distributed into pots.
(7) How was irrigation water managed?
Response 7: Watering once every 2-4 days and appropriately changed the irrigation frequency according to the actual situation.
(8) How was harvesting done?
Response 8: Harvesting completed after the end of the growth period.
(9) What cultivars were used for the melons and broccoli? Could it be possible that certain microbial communities can be favoured by certain cultivars? hence its important to disclose this. There are just too many cultivars of melons out there..
Response 9: We have added this part in the Materials and Methods.
(10) What was the experimental design? At what level of the experiment was replication of treatments done?
Response 10: We planted a total of 60 pots, and every 20 pots was a repetition.
(11) In most cases, it is only the heads of broccoli harvested and the leaves are incorporated into the soil. Was this the case in this study?
Response 11: We removed the whole plants after broccoli harvest.
Thank you again for your suggestion, I hope to learn more from you.
Reviewer 2 Report
I have had the pleasure of reviewing your manuscript titled " Effects of Broccoli Rotation on Soil Microbial Community Structure and Physicochemical Properties in Continuous Melon Cropping." I must commend your comprehensive and robust approach to this pertinent issue in the realm of agricultural science.
The issue of crop rotation is indeed a pressing one, and your research not only highlights the problem but provides a tangible and innovative solution. Your study design, especially the inclusion of a crop from the Cruciferous plants is appropriately rigorous, enabling a holistic understanding of the variables at play. Your focus on a wide array of measurements, including rhizosphere soil microbial community relative abundance and diversity during melon–broccoli rotation is highly commendable. However, there are some recommended amendments required as follow:
Point 1: The last three sentences of the introduction section, where you illustrate the main objective of the study. The hypothesis of your work needs to be more justified. Please rephrase.
Point 2: My next comment pertains to the missed data of growth characteristics. Here you did not present any measurement of growth characteristics by which are essential for your results justification. You should have done these measurements while performing the experiment. If YES, please include this data in the results. Otherwise, from the agronomical point of view, your MS fits more in a soil science journal such as Soil Systems.
Point 3: While your manuscript clearly reflects a sound understanding of the existing literature, I would like to emphasize the importance of including more recent publications in your references. Scholarly conversations are continually evolving and including citations from the last three years (especially from 2023) will ensure your work is positioned within the most current state of the field.
Point 4: Your manuscript would benefit from some minor revisions to ensure accuracy and consistency in your citations, references, figures, tables, and abbreviations.
Point 5: While your manuscript displays a strong command of the topic and presents compelling findings, I noticed some minor language issues and inconsistencies throughout the text. These could potentially hinder the clarity of your message and disrupt the reader's engagement with your work.
I noticed some minor language issues and inconsistencies throughout the text. These could potentially hinder the clarity of your message and disrupt the reader's engagement with your work.
Author Response
Response to Reviewer 2 Comments
Dear Reviewer:
We really appreciate you for your carefulness and conscientiousness. Your suggestions are really valuable and helpful for revising and improving our paper. According to your suggestions, we have made the following revisions on this manuscript:
Point 1: The last three sentences of the introduction section, where you illustrate the main objective of the study. The hypothesis of your work needs to be more justified. Please rephrase.
Response 1: I have modified this part.
Point 2: My next comment pertains to the missed data of growth characteristics. Here you did not present any measurement of growth characteristics by which are essential for your results justification. You should have done these measurements while performing the experiment. If YES, please include this data in the results. Otherwise, from the agronomical point of view, your MS fits more in a soil science journal such as Soil Systems..
Response 2: The next crop of melons had not yet been planted when we tested the soil samples.
Point 3: While your manuscript clearly reflects a sound understanding of the existing literature, I would like to emphasize the importance of including more recent publications in your references. Scholarly conversations are continually evolving and including citations from the last three years (especially from 2023) will ensure your work is positioned within the most current state of the field.
Response 3: We have added new references.
Point 4: Your manuscript would benefit from some minor revisions to ensure accuracy and consistency in your citations, references, figures, tables, and abbreviations.
Response 4: We checked the manuscript and modified some minor revisions.
Point 5: While your manuscript displays a strong command of the topic and presents compelling findings, I noticed some minor language issues and inconsistencies throughout the text. These could potentially hinder the clarity of your message and disrupt the reader's engagement with your work.
Response 5: Our manuscript has been edited in professional English on MDPI.
Thank you again for your valuable comments and suggestions. I look forward to hearing from you soon in due course.